# Melanopsin^+^RGCs Are fully Resistant to NMDA-Induced Excitotoxicity

**DOI:** 10.3390/ijms20123012

**Published:** 2019-06-20

**Authors:** Beatriz Vidal-Villegas, Johnny Di Pierdomenico, Juan A Miralles de Imperial-Ollero, Arturo Ortín-Martínez, Francisco M Nadal-Nicolás, Jose M Bernal-Garro, Nicolás Cuenca Navarro, María P Villegas-Pérez, Manuel Vidal-Sanz

**Affiliations:** 1Department of Ophthalmology, University of Murcia and Instituto Murciano de Investigación Biosanitaria (IMIB)-Virgen de la Arrixaca, 30120 Murcia, Spain; beatrizvidalvillegas@gmail.com (B.V.-V.); johnnydp@um.es (J.D.P.); juanantonio.miralles@um.es (J.A.M.d.I.-O.); Arturo.OrtinMartinez@uhnresearch.ca (A.O.-M.); fm.nadalnicolas@um.es (F.M.N.-N.); jmbg@um.es (J.M.B.-G.); mpville@um.es (M.P.V.-P.); 2Department of Physiology, Genetics and Microbiology and Multidisciplinary Institute for Environmental Studies “Ramón Margalef”, University of Alicante, 03690 Alicante, Spain; cuenca@ua.es

**Keywords:** NMDA, excitotoxicity, glaucoma, melanopsin-RGCs, intrinsically photosensitive-RGCs, Brn3a^+^RGCs, adult albino rat, retina, SD-OCT

## Abstract

We studied short- and long-term effects of intravitreal injection of *N*-methyl-d-aspartate (NMDA) on melanopsin-containing (m^+^) and non-melanopsin-containing (Brn3a^+^) retinal ganglion cells (RGCs). In adult SD-rats, the left eye received a single intravitreal injection of 5µL of 100nM NMDA. At 3 and 15 months, retinal thickness was measured in vivo using Spectral Domain-Optical Coherence Tomography (SD-OCT). Ex vivo analyses were done at 3, 7, or 14 days or 15 months after damage. Whole-mounted retinas were immunolabelled for brain-specific homeobox/POU domain protein 3A (Brn3a) and melanopsin (m), the total number of Brn3a^+^RGCs and m^+^RGCs were quantified, and their topography represented. In control retinas, the mean total numbers of Brn3a^+^RGCs and m^+^RGCs were 78,903 ± 3572 and 2358 ± 144 (mean ± SD; *n* = 10), respectively. In the NMDA injected retinas, Brn3a^+^RGCs numbers diminished to 49%, 28%, 24%, and 19%, at 3, 7, 14 days, and 15 months, respectively. There was no further loss between 7 days and 15 months. The number of immunoidentified m^+^RGCs decreased significantly at 3 days, recovered between 3 and 7 days, and were back to normal thereafter. OCT measurements revealed a significant thinning of the left retinas at 3 and 15 months. Intravitreal injections of NMDA induced within a week a rapid loss of 72% of Brn3a^+^RGCs, a transient downregulation of melanopsin expression (but not m^+^RGC death), and a thinning of the inner retinal layers.

## 1. Introduction

Light is converted by photoreceptors (rods and cones) into electrical signals, which are initially processed at the outer synaptic layer of the retina where photoreceptor information is modulated by horizontal cells and conveyed onto bipolar cells. Signals are further processed at the inner synaptic layer, where the bipolar information is modulated by amacrine cells and finally passed on to retinal ganglion cells (RGCs) in the innermost retinal layer. RGCs, the only ones whose axon leaves the retina, convey the information processed in the retina to the retinorecipient nuclei of the brain. This projection obtains relevant information from our visual world from the retina and provides it to the brain to produce image-forming as well as nonimage-forming visual functions. Retinal information that produces image-forming visual functions is carried out by the general population of RGCs that have the expression of brain-specific homeobox/POU domain protein 3A (Brn3a) in common, while the information necessary to produce nonimage-forming visual functions is carried out by a small subpopulation of RGCs that express the photopigment melanopsin (m^+^RGCs), rendering them intrinsically photosensitive (ipRGCs)—the so-called third photoreceptor cell-type of the retina [1].

In adult rodents, RGCs constitute less than 1% of all retinal cells [2,3,4]. Based on their morphology (soma size and dendritic arborization), extension of their dentritic arborization into the inner synaptic layer, electrophysiological responses to light stimulus within their receptive field, target region of the brain, and genetic background, it has been proposed that the rodent retina may have up to 40 different types of RGCs [5,6,7,8]. In rats it has been estimated that excluding endothelial cells, the ganglion cell layer is composed of approximately 50% displaced amacrine cells (ACs), 10% glial cells, and 40% RGCs [9]. Displaced ACs not only share their location in the retina with RGCs but overlap in size, thus making it difficult to distinguish RGCs from ACs, and this has obliged the use of retrogradely transported neuronal tracers [10,11] or neuronal markers to identify RGCs. There are several markers that identify large proportions of RGCS (pan-markers) or many RGC types, including Thy-1 [12], Brn3a [13,14], RBPMS [15], class III beta-tubulin [16], Neuronal Nuclei (NeuN) [17], and Microtubule-associated protein 1A (MAP 1A) [7,18]. In addition, there are several markers that allow identification of specific types of RGCs, such as melanopsin [19] and others [7,8,20]. However, after retinal injury, many of the physiological and morphological attributes of RGCs, including their dendritic arborization, may change [8,21,22], and the molecular markers may be downregulated, rendering the identification of RGCs difficult [23,24,25,26,27,28].

The characterization of the expression of Brn3a by rodent RGCs has allowed identification of the main population of RGCs that convey image-forming visual information to the brain, which represents approximately 96% of the RGC population [14]. Nonimage-forming visual behaviours depend on intrinsically photosensitive RGCs (ipRGCs), one type of RGC with a large dendritic arbor that contains the photopigment melanopsin (m^+^RGCs), which is responsible for the circadian photoentrainment, pupillary reflexes, and the regulation of pineal melatonin secretion [1,29,30]. Six subtypes of ipRGCs have been described to express at least small amounts of melanopsin (also known as Opn4), and are named M1-M6 [31,32]. Antibodies against melanopsin allow the identification of the large majority of ipRGCs, preferentially M1–M3, because M4 (which corresponds to the ON αRGC subtype [33,34]), M5 [35], and M6 [32] express less Opn4 than M1–M3 and are difficult to identify with standard immunohistochemistry [31,32,36,37,38,39,40]. In rats, the population of m^+^RGCs constitute approximately 2.5 and 2.7% of the pigmented and albino RGC populations, respectively [13,14,19,41,42]. Moreover, because Brn3a and melanopsin are rarely expressed in the same RGC, immunohistofluorescent studies using these two markers together allows the study, in parallel but independently, of the responses of these two types of RGCs to different retinal injuries [28,43].

Glutamate excitotoxicity may be induced by the intravitreal injection of *N*-methyl d-Aspartate (NMDA), which results in the excessive stimulation of NMDA receptors, one of the three ionotropic glutamate receptor subtypes widely expressed by inner retinal neurons. Glutamate excitotoxicity is thought to play an important role in the loss of RGCs in various retinal injuries [44,45], including glaucoma [46,47,48,49,50], transient ischemia [51], and optic nerve injury [52,53], and may also play a key role in many CNS diseases involving neuronal death [54]. Excessive NMDA receptor stimulation may result in alterations of the Na^+^/K^+^ homeostasis, excessive influx of large amounts of Ca^2+^ into the cell [55], which may result in direct damage by activation of enzymes that damage DNA and cell membranes [56] and by the induction of apoptosis through activation of c-AMP [57]. Animal models of NMDA-induced retinal excitotoxicity are often used to explore molecular mechanisms of RGC apoptosis and its protection [58,59,60,61,62,63,64,65,66].

The susceptibility of RGCs to NMDA-mediated excitotoxicity has been studied previously in adult rats [58,62] and mice [59,67], as well as the effects of intravitreal NMDA on the specific type population of m^+^RGCs [59,67]. However, these were short term studies spanning up to 58 days after NMDA injection, and thus the short- and long-term effects of NMDA excitotoxicity on the population of RGCs expressing Brn3a have not been investigated so far. Moreover, to what extent NMDA-induced neurotoxicity may result in long term effects on the retinal architecture and on the population of ipRGCs itself have not been previously investigated.

In the present study we take advantage of recent techniques developed in the laboratory to identify, count, and map in the same retinal wholemounts the populations of RGCs expressing Brn3a or melanopsin. Moreover, we use modern non-invasive techniques, such as the Spectral Domain Optical Coherence Tomography (SD-OCT), to image and analyze retinal thickness longitudinally at short (3 months) and long (15 months) survival intervals. We investigate the responses of the general population of RGCs (Brn3a^+^) and the population of ipRGCs (m^+^RGCs) to excitotoxicity induced by the intravitreal injection of NMDA. Overall, our studies indicate that the general population of Brn3a^+^RGCs is quite sensitive to NMDA-mediated excitotoxicity, which in one week induces the loss of approximately 72% of the population. In contrast, m^+^RGCs, after a transient downregulation of melanopsin, show a remarkable capacity for survival of the entire m^+^RGC population for periods of up to 15 months. Examination of these retinas with SD-OCT reveals that NMDA-injected retinas showed a marked reduction in the thickness of the total and inner retina that was present at 3 months and progressed up to 15 months. Short accounts of this work have been published in abstract format [68].

## 2. Results

We have included in this study a total of 51 rats whose left eye received an intraocular injection of 5 µL NMDA (100 nM). The first 28 were analyzed within the first 14 days after the injection while the remaining 23 were analyzed at 15 months to investigate the long-term effects of the excitotoxic insult on the survival of two RGC populations—the Brn3a^+^RGCs and the m^+^RGCs. Five additional naïve rats were used as controls. In addition, SD-OCT was used to measure retinal thickness in both retinas of 18 animals at 3 and 15 months after NMDA injection.

### 2.1. Rapid and Massive Loss of Brn3a^+^RGCs Shortly after NMDA Injection

When the right or naïve retinas were examined under a fluorescence microscope, Brn3a^+^RGCs showed typical distribution throughout the entire retina with higher densities on the superior retina, just above the optic nerve along the visual streak, as described in detail previously [69,70,71]. Changing the fluorescent filter allowed us to see m^+^RGCs distributed in a complementary fashion to Brn3a^+^RGCs, and as previously shown by this laboratory [14,19], we were not able to see any doubly immunolabelled RGC, thus confirming that these markers are exclusive to one population (Figure 1).

Total numbers of Brn3a^+^RGCs (78,903 ± 3573 mean ± SD, *n* = 10) in the naïve retinas were comparable to those in the right fellow retinas of our experimental groups analyzed at 3, 7, and 14 days (76,472 ± 5815 Brn3a^+^RGCs mean ± SD, *n* = 29), or 15 months (81,480 ± 5602 mean ± SD, *n* = 20) after NMDA injection, as well as to those obtained in previous studies from this laboratory [13,14,19,72] (Figure 1 and Figure 2, Table 1).

The left NMDA-injected retinas showed significant decreases in the total numbers of Brn3a^+^RGCs. By 3 days after NMDA injection, the total number of Brn3a^+^RGCs was 38,940 ± 22,443 (*n* = 9), which is significantly smaller than naïve controls and contralateral retinas (Mann Whitney test, *p* ≤ 0.001). There were further reductions at 7 (21,811 ± 9750 mean ± SD, *n* = 6) and 14 days (19,348 ± 8502 mean ± SD, *n* = 10) but these were not statistically significant when compared to 3 days, indicating that in this injury model RGC loss occurs early after NMDA injection, but there is no further progression between 3 and 14 days (Figure 2, Table 1). Moreover, at 15 months, the left NMDA-injected retinas showed significantly lower numbers than their fellow retinas (15,099 ± 8595 mean ± SD, *n* = 23) that corresponded to a survival of approximately 19%, and these values were significantly smaller than those observed at 3 days (Mann Whitney test, *p* = 0.019), but not from those obtained at 7 days (Mann Whitney test, *p* = 0.187), indicating that there is no further loss of Brn3a^+^RGCs between 7 days and 15 months (Figure 1, Figure 2 and Figure 3, Table 1).

Retinal distribution of Brn3a^+^RGCs in the NMDA-injected retinas did not adopt any particular spatial pattern; their loss was diffuse and distributed over the entire amount of retinas (Figure 3), although occasionally there was a smaller density in the superior temporal quadrant that could be explained by the proximity to the intraocular puncture, and thus, a region exposed to a greater concentration of the injected NMDA.

### 2.2. After A Transient Downregulation of Melanopsin, m^+^RGCs Appear Fully Resistant to NMDA Injection

Total numbers of m^+^RGCs (2358 ± 143 mean ± SD, *n* = 10) in the naïve retinas were comparable to those obtained in the right fellow retinas of our experimental groups analyzed at 3, 7, and 14 days (2257 ± 228 m^+^RGCs mean ± SD, *n* = 29), or at 15 months (2166 ± 96 mean ± SD, *n* = 9) after NMDA injection, as well as to those obtained in previous studies from this laboratory [13,19,69] (Figure 1, Figure 3 and Figure 4, Table 2).

By 3 days after intravitreal injection of NMDA, the total number of m^+^RGCs was 1516 ± 312 (*n* = 10), a significant reduction when compared to naïve or contralateral retinas (Kruskal Wallis test, *p* ≤ 0.001) (Figure 2). Surprisingly, the total number of m^+^RGCs at 7 or 14 days after NMDA injection was 2105 ± 445 (*n* = 7) or 2419 ± 257 (*n* = 11), showing a significant increase when compared to the values observed at 3 days, and reached comparable values to those of control retinas by 14 days (Kruskal Wallis test, *p* > 0.05). By 15 months after NMDA injection, the left retinas showed a total number of m^+^RGCs (2027 ± 134 mean ± SD, *n* = 11) comparable with the data obtained in their right fellow retinas (2166 ± 96 mean ± SD, *n* = 9) (Mann Whitney test, *p* = 0.518).

We interpret this abrupt decrease and subsequent recovery of the total number of m^+^RGCs as a transient downregulation of melanopsin, shortly after intravitreal injection of NMDA, which recovers up to normal levels of expression and total number of m^+^RGCs by 7 days, 14 days, and 15 months. In addition, these results also indicate that m^+^RGCs are resistant to NMDA-induced excitotoxicity. In contrast with the Brn3a^+^RGC population, whose total numbers were reduced to approximately one quarter to one fifth of their normal values, the m^+^RGCs show a complete population that is comparable to that found in their fellow contralateral and in naïve retinas (Figure 1, Figure 3 and Figure 4, Table 2).

### 2.3. In Vivo SD-OCT Measurements

We wanted to examine the effects of the NMDA-induced retinal degeneration on the retinal layers, and thus retinas were analyzed at 3 and 15 months with SD-OCT to determine the total and inner retinal thickness. Figure 5 shows representative SD-OCT images from both eyes in two representative experimental rats analyzed longitudinally in vivo 3- and 15-months after NMDA-injection. The SD-OCT provided measurements of the 31 sections acquired, and we selected three sections located superior, central, or inferior for analysis. Because the measurements of these three sections were comparable within each individual retina and time interval examined, the values from these 3 sections were pooled and used as a value for each retina and time point.

Total retinal (TR) thickness (as measured in µm from the inner side of the nerve fiber layer to the outer limit of the outer segment layer) was significantly smaller in the NMDA-injected retinas as compared to their contralateral fellow retinas at 3 months (185 ± 4 versus 211 ± 3.9; *n* = 18) and 15 (162 ± 6.1 versus 195 ± 6.1; *n* = 18) months. In fact, the thinning of the TR was mainly due to the thinning of the inner retina (IR) (as measured in µm from the inner side of the nerve fiber layer to the outer limit of the inner nuclear layer). The IR thickness in the left NMDA-injected eyes was significantly smaller than in their fellow retinas at 3 months (83 ± 3.7 versus 96 ± 2.6; *n* = 18) and 15 (71 ± 2.8 versus 91 ± 3.4; *n* = 18) months (Figure 5 and Figure 6).

The TR thickness of the fellow retinas diminished significantly between 3 (211 ± 3.2; *n* = 18) and 15 (195 ± 6.1; *n* = 18) months, a finding that is in agreement with recent studies in adult albino rats showing a physiological thinning of the TR and IR of approximately 16 and 6 µm, respectively, with age [69]. However, superimposed to the physiological age-related thinning of the retina, in the experimental NMDA-injected retinas there was further significant thinning of the TR (23 µm) and IR (12 µm) between 3 and 15 months (Figure 5 and Figure 6).

## 3. Discussion

Here we have investigated the short- and long-term responses of the populations of Brn3a^+^ and melanopsin expressing (m^+^) RGCs after an excitotoxic insult to the retina. Our studies show that following an intraocular injection of 100 nM NMDA, there is a rapid and massive loss of the general population of Brn3a^+^RGCs; by 3 days, 14 days, or 15 months, the surviving population represents approximately 49%, 24%, or 19%, respectively, of the original population. When examined with SD-OCT there was an important reduction in the thickness of the total and the inner retina at 3 months that further progressed up to 15 months. 

Compared to the population of Brn3a^+^RGCs, m^+^RGCs show by 3 days a transient downregulation of melanopsin that recovers over the next weeks, and by 14 days or 15 months the numbers of m^+^RGCs are comparable to their contralateral fellow eyes.

When studying the responses of RGCs to retinal injuries it is important to be able to identify different types of RGCs to understand how these respond to injury [43,73]. Here, we use modern techniques developed in the laboratory to count, image, and represent the retinal topography of two RGC populations that can be readily identified with Brn3a and melanopsin [43,74]. Recent studies from this laboratory have demonstrated that in the adult rat, retinal injuries induce a transient downregulation of melanopsin [28], followed by the expression of melanopsin in injured neurons surviving long periods of time [9,19,75,76]. Of the six main subtypes of ipRGCs M1–M6, immunocytochemistry against melanopsin identifies mainly M1–M3 because they show higher levels of melanopsin expression [32,33,34,35,37,38], and thus when interpreting our data, we should take into account that our immunohistochemical methods primarily identify the M1–M3 ipRGC subtypes. In fact, although not analyzed in this work, it is conceivable that most of our results refer to the M1 and M2 subtypes, which are the most abundant and readily identified with melanopsin antibodies [37,38,40,77].

### 3.1. Intravitreal Injection of NMDA Induces Brn3a^+^RGC Death

The loss of RGCs observed after the injection of NMDA in our studies is comparable to that found by others in mice [59,67] or rats [53,58,78] analyzed at survival intervals ranging 3–58 days. We noticed certain inter-animal variability in the total number of surviving Brn3a^+^RGCs at 3 days after NMDA injection, which was also reported by others [58,67] and could be due to an individual animal susceptibility, or to the fact that RGC loss has not concluded by that time interval. Inter-animal variability following other types of retinal injuries, such as intraorbital optic nerve cut or crush, an insult that results in axotomy of the entire RGC population, have been shown [9,79]. Another possible explanation for the inter-animal variability could be the fact that intravitreal injections may suffer a small reflux of the injected volume rendering the concentration of NMDA unequal for all eyes. We have not investigated shorter survival intervals than 3-days after NMDA injection, but other studies have suggested that following NMDA injection RGC loss appears as early as 6-h after injection [80]. It is currently thought that NMDA-induced excitotoxicity results in activation of the NMDA receptor and this leads to a massive influx of Ca^++^ that acts as a second messenger to activate pathways that lead to apoptotic neuronal death [81], although the exact signalling pathways involved in NMDA-induced RGC death are not completely understood [61].

By 14 days, approximately 75% of the original Brn3a^+^RGC population has degenerated. Their topography does not reveal any particular distribution, such as the patchy or sectorial patterns observed after acute or chronic ocular hypertension, respectively [82,83,84], but rather a diffuse pattern of RGC loss much like the pattern observed after crushing or cutting of the intraorbital optic nerve [70,74]. In some instances, over the general diffuse pattern of RGC loss, there was a marked decrease in the superior temporal quadrant, the region where NMDA was injected, and thus where NMDA may have reached a higher concentration.

Our results indicate that RGC loss is not progressive, since between 7-days and 15-months after NMDA injection there are no significant changes in total numbers of Brn3a^+^RGCs. Our results appear in disagreement with those of Huang and colleagues [78] that reported a progressive RGC loss between 12 h and 58 days, but these authors employed Neu-N antibodies as a marker to identify RGCs, which identify displaced amacrine cells in addition to RGCs [17], and RGC densities were obtained by averaging 16 samples obtained throughout the retina, whereas our numbers were obtained after counting the entire amount of retinas. Nevertheless, Huang and colleagues [78] reported the loss of approximately 67% of the RGC population at 58 days, which is analogous to the 75% loss observed in our studies.

### 3.2. Intravitreal Injection of NMDA Induces A Progressive Retinal Thinning

RGC degeneration results in the loss of neural processes that extend into the inner synaptic layer, where they contact cone-bipolar and amacrine cells of different types forming an extensive neuropil that makes up a substantial proportion of the inner synaptic layer’s volume. Our results indicate that NMDA-induced retinal excitotoxicity results in a significant decrease of the total (TR) and inner (IR) retinal thickness. This thinning was already apparent in the left NMDA-injected experimental retinas at 3 months when compared to their fellow retinas. The retinal thinning may be explained because over 75% of the Brn3a^+^RGC population is missing and their dendrites have degenerated, thus prompting a thinning of the IPL [78], but also because NMDA-excitotoxiticy results in loss of amacrine cells, as shown with Terminal deoxynucleotidyl transferase dUTP nick end labeling (TUNEL) and morphometric techniques in adult pigmented mice [85,86,87] and albino rats [78,88]. The thinning of the TR and IR observed in the fellow retinas between 3 and 15 months is consistent with the physiological thinning of the adult SD rat retina with age [72]. However, superimposed on this physiological thinning, in the experimental retinas there was a progressive thinning of the TR and IR between 3 and 15 months, indicating a continuing retinal degeneration prolonged beyond the time of NMDA injection and the period of Brn3a^+^RGC loss, which concluded at 7 days after the injection. A possible explanation for the progressive thinning of the IR could be the secondary amacrine cell loss that follows RGC death observed after NMDA-induced neurotoxicity. Indeed, approximately 72% of the RGC types in the mice retina are coupled to ACs [89], which may possibly facilitate secondary cell loss of calretinin, calbindin, and choline acetyltransferase immunopositive ACs via gap junctions [87].

### 3.3. The m^+^RGCs Resilience to Retinal Disease and Injury

In the adult rat, m^+^RGCs only represent approximately 2.7% or 2.5% of the total population of RGCS in albino or pigmented populations, respectively [14,19,72]. Yet, the availability of specific molecular markers for this type of RGCs has made it possible to learn in a very short period of time a great deal about the morphological and functional properties of these neurons, including their idiosyncratic response to different types of inherited or acquired retinal lesions [43]. A number of different laboratories have shown that ipRGCs demonstrate a much better survival against a variety of retinal injuries than the general population of RGCs [90], and this particular resilience has been shown against ocular hypertension in rats [39,91] or mice [92], crushing or cutting of the optic nerve in rats [93,94] or mice [35,76,95,96], and transient ischemia of the retina in rats [84]. However, ipRGCs do not appear to be particularly resilient in inherited models of retinal degeneration [95,96,97,98,99], mitochondrial optic neuropathies [100], or degenerative diseases [77], such as Alzheimer’s [101], Parkinson’s [102], or Huntington’s [103] disease. A detailed characterization of the RGC responses to NMDA-induced excitotoxicity may shed light into the paradigm of the different responses of different populations of RGCs to injury; why some populations die while others survive.

### 3.4. The m^+^RGCs Are Resistant to NMDA-Induced Retinal Excitotoxicity

Our results demonstrate that following a transient downregulation of melanopsin expression, the total number of m^+^RGCs at 14 days or 15 months is comparable to their contralateral fellow eyes, thus indicating an outstanding endurance to NMDA-induced excitotoxicity.

Survival of the entire m^+^RGC population at 15-months after NMDA injection is underscored in view of the important inner retinal degeneration and loss of approximately 81% of the Brn3a^+^RGC population. The degeneration of RGCs following NMDA-induced excitotoxicity has been explored in adult pigmented mice analyzed at 6 [67] or from 1 to 21 [59] days, respectively. However, these studies showed slight differences in terms of the survival of the m^+^RGC population. DeParis and colleagues [67] found that 6 days after NMDA injection there was a full component of m^+^RGC population surviving in the retina with no downregulation of the expression of melanopsin, while Wang and colleagues [59] reported the loss of approximately one half of the m^+^RGC population at 21-days after NMDA injection. These differences may be explained by the diverse amount of NMDA injected (3 µL of 10 mM NMDA versus 1 µL of 40 mM NMDA).

The downregulation of melanopsin expression that occurs after retinal injury requires further consideration. Our studies reveal that following NMDA injection there is a transient downregulation of melanopsin that recovers fully by 14 days. A similar transient downregulation of melanopsin has been described in previous studies from this laboratory in adult rats following optic nerve injury [94], transient elevation of the intraocular pressure [84], the use of retrogradely transported neuronal tracers [104], or acute light-induced retinal degeneration [75]. The differences between our results and those observed by DeParis and colleagues [67] may be a species-specific response of m^+^RGCs, because in parallel studies of m^+^RGC survival in adult mice following intraorbital optic nerve injury, we did not find a transient downregulation of melanopsin [28,76].

Of all the retinal injuries examined so far, m^+^RGCs best tolerate NMDA-induced excitotoxicity. The reasons for the remarkable resilience of ipRGCs to survive different types of injury-induced retinal degeneration remains an open issue for future studies, but several hypotheses have been forwarded to explain m^+^RGC resilience. One hypothesis proposes that these ipRGCs have large dendrites within the inner synaptic layer, and thus their intra-retinal connections may be enough to provide trophic support for survival in the absence of brain-target-derived trophic support [29,90,93,105]. Although it has been postulated that the absence of NMDA receptors in m^+^RGCs could explain their particular resistance to NMDA-mediated excitotoxicity, it has been shown that all RGCs express NMDA receptors [106], including m^+^RGCs [66,107], and that the particular resilience of m^+^RGCs is not related to pigmentation, genetic background, the presence of photoreceptors, or the activation of the endogenous survival JAK/STAT pathway [67]. Other possible explanations include the activation of the PI3K/AKT pathway after cutting of the optic nerve or ocular hypertension [108], but this was not apparent in NMDA-induced excitotoxicity [67]. Melanopsin itself could be thought to have an effect on cell survival, but the fact that many ipRGCs survive with a transient, but lower, expression of melanopsin makes this unlikely. Another hypothesis explains the resilience on the basis of their neurotransmitter (PACAP) and it is hypothesized that PACAP would act as a neuroprotectant conferring these neurons their particular resistance, since exogenous administration of PACAP protects RGCs against optic nerve transection [109], ocular hypertension [110], or NMDA administration [111]. It could also be possible that different types of RGCs may have different responses to the same insult, thus arguing in favour of a type-specific susceptibility. For example, recent studies using genetic markers to identify different types of RGCs have shown that the type of αRGC is particularly resistant to NMDA induced neurotoxicity [8] or to optic nerve crush [35,39], in contrast to the very low survival of junction adhesion molecules B-expressing RGCs (J-RGCs) [8]. Moreover, recent studies indicate that different subtypes of ipRGCs have different susceptibility to specific insults; for instance, in a mouse model of Huntington’s disease (HD), M1 were reduced compared to non-M1 ipRGCs that survived to HD progression [112]. Furthermore, in a mouse model of ocular hypertension, subtypes of αRGCs were found to have different susceptibility, with OFF-transient αRGCs being more vulnerable than ON- or OFF-sustained αRGCs [22,73]. Overall, the particular resilience of m^+^RGCs makes them a suitable candidate to study changes in protein expression after injury, which furthers our knowledge about what makes a neuron survive better than others, and this could in turn result in the design of new neuroprotective strategies for RGCs against noxious stimuli. Thus, future studies are needed to decipher the molecular correlations that provide these neurons with a self-built neuroprotection against various types of injury, including NMDA-induced RGC death.

## 4. Material and Methods

### 4.1. Animal Handling and Experimental Groups

Experiments were prepared in 56 adult female SD rats (250 g) obtained from the animal house (Murcia University, Murcia, Spain) and treated according to the European Union guidelines for Animal Care and use of scientific purpose (Directive 2010/63/UE). All procedures were approved by the Ethical and Animal Studies Committee of the University of Murcia, Spain (Comité Ético de Experimentación Animal (CEEA) de la Universidad de Murcia, 11 January 2017, Code: A13170110 and A13170111). Animals had free access to food and water and were kept in a temperature and light controlled room with 12-h/12-h light/dark cycles. Animals were anaesthetized with a mixture of xylazine (10 mg/kg Rompun; Bayer, Kiel, Germany) and ketamine (60 mg/Kg bw, Ketolar; Pfizer, Alcobendas, Madrid, Spain); 0.5% proparacaine hydrochloride eye drops (Alcon Co., Fort Worth, TX, USA) were used to achieve topical anaesthesia. After the surgical procedures, an ocular ointment was placed over the corneas of both eyes to prevent corneal desiccation (Tobrex^®^; Alcon-Cusí, S.A., Barcelona, Spain). Animals were divided into experimental and control groups. The experimental group received an intraocular injection of NMDA and was divided into four subgroups that were examined at 3 (*n* = 10), 7 (*n* = 7), or 14 (*n* = 11) days, or 15 months (*n* = 23). Additional naïve rats (*n* = 5) were used as controls. For animal sacrifice an overdose of sodium pentobarbital was injected intraperitoneally (Dolethal, Vetoquinol^®^, Especialidades Veterinarias, S.A., Madrid, Spain).

### 4.2. Intraocular Injections of NMDA

Retinal excitotoxicity was induced in the left eye of the experimental animals by intraocular injection of 5 µL of 100nM NMDA *N*-methyl-d-Aspartate (NMDA) (M3262; Sigma-Aldrich Química S.A., Madrid, Spain) dissolved in 0.1 M phosphate buffer saline (PBS) following standard techniques in our laboratory [113,114,115]. In brief, a small puncture in the sclera approximately 1 mm from the limbus was made with a 30-gauge needle, and then NMDA was injected slowly with a Hamilton syringe whose needle was introduced through the sclerotomy. After injection, the needle was withdrawn slowly and an ointment (Tobrex pomada; Alcon S.A., Barcelona, Spain) was placed over the eyes to prevent corneal dehydration until anaesthesia recovery. The contralateral non-injected eye was used as control, and 5 naïve rats (10 eyes) were also used as controls. Preliminary experiments allowed us to try increasing doses of NMDA to find one that would result in consistent RGC death. Previous studies from this Laboratory did not find any effect of the intraocular injection of vehicle alone (0.1 M phosphate buffer saline, PBS) on the survival of the Brn3a^+^ or melanopsin^+^ RGC populations, and thus, we did not employ additional animals for this purpose.

### 4.3. In Vivo Measurements of the Retinal Thickness with SD-OCT

SD-OCT measurements were obtained to analyze changes in the thickness of the retina following NMDA intraocular injection, and the eyes were imaged at 3 and 15 months, as previously described in detail [72,116]. Rats were anaesthetized systemically, and eye drops were placed on both eyes to induce mydriasis (Tropicamida 1%; Alcon-Cusí, S.A.) and to prevent corneal desiccation (artificial tears). Rats were placed in prone position over a platform with their heads upright and turned to the opposite side of the inspected eye. The head position was kept similar for all animals, and for the following examination the follow up tool of the OCT program was used to compare the same regions. Both retinas were examined with OCT following manufacturer guidelines (Spectralis; Heidelberg Engineering, Heidelberg, Germany). To compensate for the rat’s corneal curvature and to maintain its hydration, we placed over the cornea a customized permeable contact lens (3.5-mm posterior radius of curvature, 5.0-mm optical zone diameter, 5.0-diopter back vertex power). To observe the rat’s eye fundus we placed in front of the camera a commercially available 78-diopter double aspheric lens (Volk Optical, Inc., Mentor, OH, USA). Photographs were taken with a software package (EyeExplorer, version 3.2.1.0; Heidelberg Engineering). Retinal thickness was determined with a raster scan of 31 equally spaced horizontal B-scans (3000 µm length) centred on the optic nerve head. For each section total retinal (TR) (as measured from the inner limiting membrane to the outer limit of the pigmented epithelial layer) and inner retinal (IR) (as measured from the inner limiting membrane to the outer limit of the inner nuclear layer) thickness were measured at distances of 1800 µm from optic disc. A total of 18 rats were analyzed longitudinally at 3 and 15 months.

### 4.4. Retinal Dissection, Immunohistochemistry and Image Acquisition

At different survival intervals, rats were sacrificed and perfused through the heart, first and briefly with a solution of 0.9% NaCl, and then slowly with a 4% paraformaldehyde solution in PBS. The superior pole of the eye was marked with a small suture, and retinas were then dissected and prepared as flattened wholemounts, as previously described [117]. Retinas were double-immunodetected following previously described methods for Brn3a and melanopsin to identify surviving RGCs expressing these two markers [14]. Primary antibodies were goat anti-Brn3a (1:750 dilution, C-20 Santa Cruz Biothechology, Heidelberg, Germany) and rabbit anti melanopsin (1:500 dilution, PAI-780, Invitrogen, Thermo Fisher Scientific, Alcobendas, Madrid, Spain). Secondary antibodies were Alexa Fluor conjugated (donkey anti-rabbit Alexa 594, donkey anti-goat Alexa 488) (Molecular Probes Thermo-Fisher, Madrid, Spain). At the end of the immunohistochemical procedure, both retinas were mounted (vitreal side up) on gelatinized slides with an antifading solution [14].

Using epifluorescence microscopy (Axioscop 2 Plus; Zeiss Mikroscopie, Jena, Germany) retinas were photographed according to standard methods in the laboratory [76,118]. A total of 154 frames were obtained in the microscope to reconstruct the whole retina. These reconstructions were obtained under both filters to allow identification of Brna3a^+^RGCs and m^+^RGCs, respectively. Following standard procedures in the laboratory [74,119,120], wholemount reconstructions were further processed to automatically obtain the total number of Brn3a^+^RGCs and their topographical distribution was represented as isodensity maps. For the m^+^RGCs, these were quantified manually and dotted on the photomontage with the aid of a graphic editing software Adobe Photoshop CS8.01 (Adobe Systems, Inc., San Jose, CA, USA). Dots were automatically identified, and their topographical distribution was represented as neighbour maps following previously described methods [120].

### 4.5. Statistics

All data is expressed as means ± standard deviation (SD). Statistical analysis employed the program GraphPad Prism^®^ for Windows (Version 5.01; GraphPad Software Inc., La Jolla, CA, EEUU) using non-parametric tests (Kruskal Wallis and Mann Whitney). Differences were considered significant if *p* < 0.05.

## 5. Conclusions

Intravitreally administered NMDA in adult albino rats: (i) induces a massive diffuse loss of Brn3^+^RGCs, which is evident within the first week and does not progress further; (ii) causes a thinning of the inner retina at 3 months that further progresses up to 15 months; (iii) triggers a transient downregulation of melanopsin expression that is evident at 3 days and recovers fully by 14 days, and; (iv) does not induce m^+^RGC loss.

## Figures and Tables

**Figure 1 ijms-20-03012-f001:**
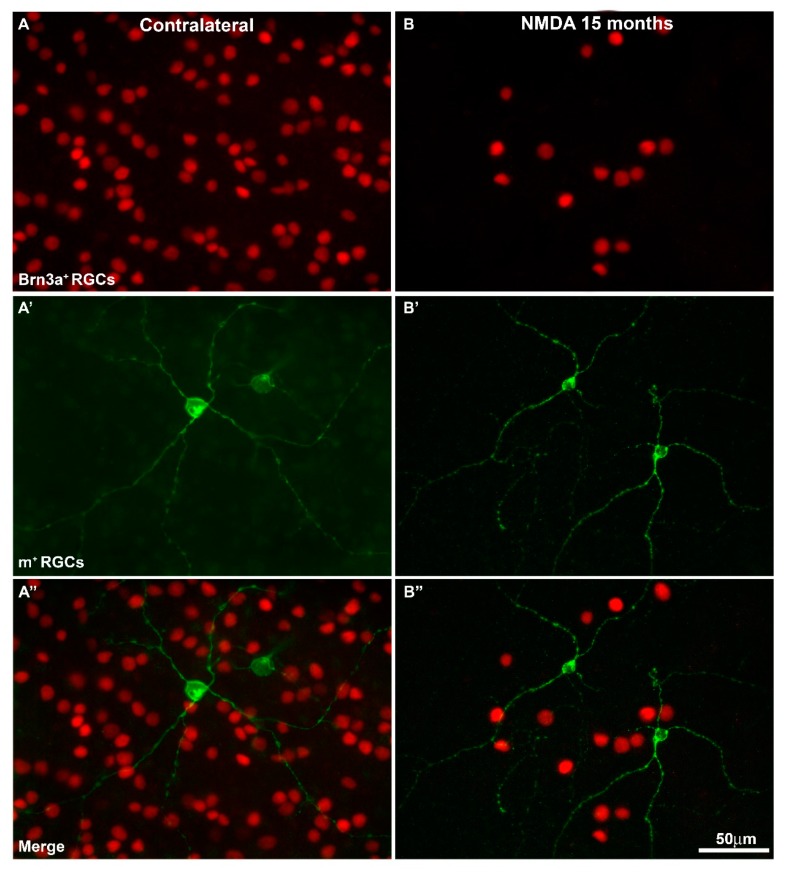
Magnifications from flat mounted retinas showing brain-specific homeobox/POU domain protein 3A positive retinal ganglion cells Brn3a^+^RGCs (**A**,**B**) and melanopsin positive retinal ganglion cells m^+^RGCs (**A’**,**B’**), and both signals (merge) (**A’’**,**B’’**) in contralateral (**A**,**A’’**) and *N*-methyl-d-aspartate (NMDA)-treated retinas (**B**,**B’’**) analyzed at 15 months after the injection. Brn3a labels cell nuclei, while melanopsin allows one to see cell somata as well as primary dendrites on the plane of focus. When both images are overlapped (**A’’**,**B’’**), one can appreciate the smaller density of m^+^RGCs compared to Brn3a^+^RGCs, as well as the fact that there are no doubly labelled RGCs. Note that 15 months after NMDA injection there are fewer Brn3a^+^RGCs. Scale bar = 50 µm.

**Figure 2 ijms-20-03012-f002:**
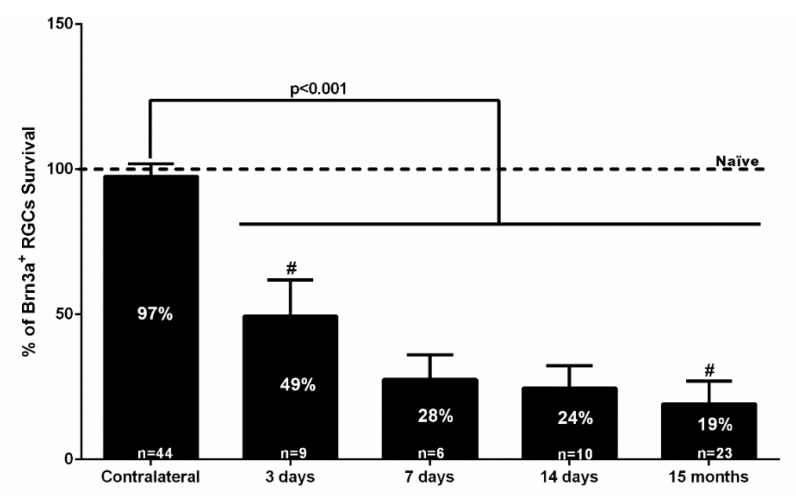
Bar graph showing the percent vs. intact retinas of the total numbers of Brn3a^+^RGCs ± standard deviation quantified in the contralateral uninjured and experimental retinas analyzed 3, 7, 14 days, or 15 months after the intraocular injection of 100 nM NMDA. The number of Brn3a^+^RGCs in the intact naïve retinas was considered 100%. The number of analyzed retinas is shown at the bottom of each bar. Statistically significant differences were observed (Kruskal-Wallis test, *p* < 0.001) between values obtained in intact retinas (naïve) or right eye retinas (contralateral) and experimental retinas examined at 3, 7, 14 days, or 15 months. When compared with the previous time study interval (at 7 days, 14 days, or 15 months), there were no significant differences (Kruskal-Wallis test, *p* > 0.05). However, there were significant differences between 3 days and 15 months (**^#^** Mann Whitney test, *p* = 0.019), but not from those obtained at 7 days (Mann Whitney test, *p* = 0.187), which suggests that NMDA-induced Brn3a^+^RGC loss does not progress between 7 days and 15 months.

**Figure 3 ijms-20-03012-f003:**
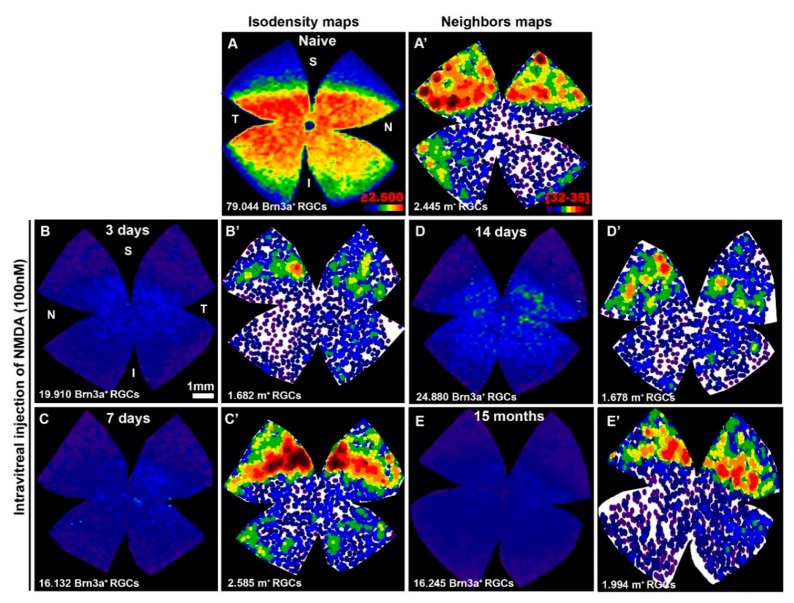
(**A**–**E**) Isodensity maps showing the retinal topography of Brn3a^+^RGCs in intact retinas (**A**) or in representative retinas analyzed at 3 (**B**), 7 (**C**), 14 (**D**) days, or 15 (**E**) months after intravitreal injection of 100 nM NMDA. (**A’**–**E’**) Neighbor maps illustrating the distribution of m^+^RGCs in the same retinas shown in (**A**–**E**). Isodensity maps color scale ranges from 0 (purple) to ≥ 2500 (red) cells/mm^2^. Neighbor map color scale, where each color represents an increase of 4 neighbors in a radius of 0.22 mm from purple (0–4 neighbors) to dark red (32–35 neighbors). Below each map is shown the total number of Brn3a^+^RGCs or m^+^RGCs counted. Note: S = superior; I = inferior; N = nasal; T = temporal. Scale bar = 1 mm.

**Figure 4 ijms-20-03012-f004:**
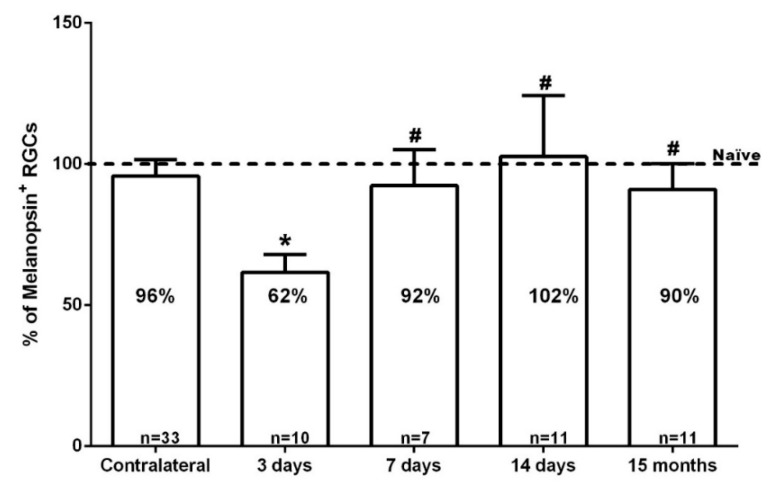
Bar graph showing the percent vs. intact retinas of the total number of m^+^RGCs ± standard deviation quantified in the contralateral uninjured and experimental retinas analyzed 3, 7, 14 days, or 15 months after the intraocular injection of 100 nM NMDA. The number of analyzed retinas is shown at the bottom of each bar. Note: * Significant differences compared to naïve, contralateral retinas, and other time points (Kruskal-Wallis test, *p* < 0.001); ^#^ the percent of m^+^RGCs in the experimental groups analyzed at 7 days, 14 days, or 15 months did not differ significantly from their contralateral fellow retinas (Mann-Whitney Test, *p* > 0.05).

**Figure 5 ijms-20-03012-f005:**
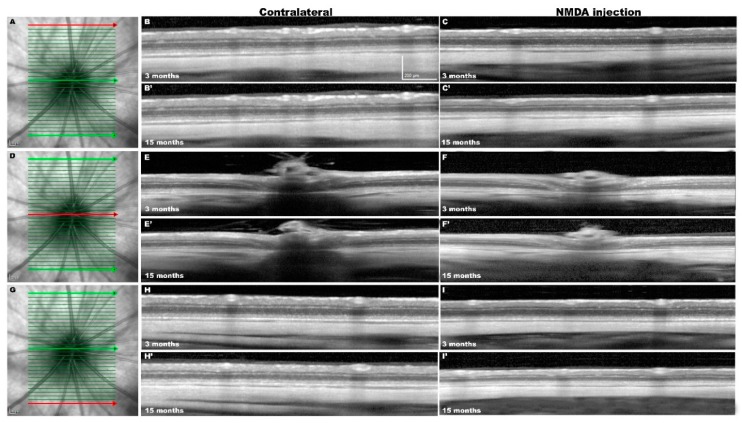
In vivo Spectral Domain-Optical Coherence Tomography (SD-OCT) images from the same contralateral and experimental retinas analyzed 3 and 15 months after NMDA injection. (**A**,**D**,**G**) Representative images of the ocular fundus of the contralateral retina and position of the 31 sections acquired. The superior (**A**), central (**D**), or inferior (**G**) retinal sections are marked in red. (**B**,**C**,**E**,**F**,**H**,**I**) Representative sections acquired (in red) from SD-OCT volume raster scan in contralateral (**B**,**E**,**H**) and NMDA-injected (**C**,**F**,**I**) retinas examined longitudinally at 3-months (**B**–**I**) and at 15-months (**B’**–**I’**) after NMDA injection.

**Figure 6 ijms-20-03012-f006:**
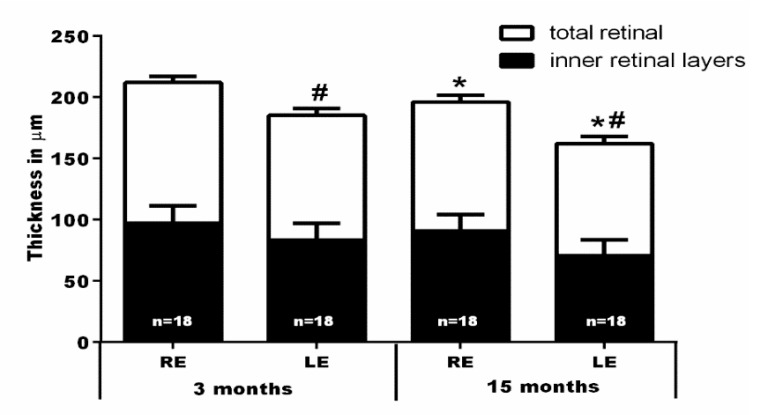
Graph bars showing the reduction of the mean ± SD thickness (µm) of the total (from inner side of the nerve fiber layer to outer segment layer) and inner (from the inner side of the nerve fiber layer to outer margin of inner nuclear layer) retina after NMDA intravireal injection into the left eye, measured in the volume scan analyses shown in Figure 5. Note: * Significant differences compared to the same eyes analyzed at 3 months (Kruskal-Wallis test, *p* < 0.001); ^#^ significant differences when compared to their contralateral eyes at the same time interval (Mann-Whitney Test, *p* < 0.001); RE = right fellow eye; LE = left eye (NMDA-injected).

**Table 1 ijms-20-03012-t001:** Total numbers of Brn3a^+^RGCs in right (RE) or left (LE) eye retinas.

Retinas	Naïve		3 Days		7 Days		14 Days		15 Months	
	RE	LE	RE	LE	RE	LE	RE	LE	RE	LE
1	80,293	82,587	72,071	46,569	74,963	24,880	71,159	16,434	89,717	14,852
2	80,399	79,044	79,209	52,957	77,604	12,227	80,940	13,785	93,939	9538
3	78,344	71,826	78,178		72,411	33,105	78,786	10,593	88,081	24,936
4	74,865	77,395	79,256	19,910	66,564		73,895	39,166	81,353	21,955
5	84,031	80,247	82,406	15,648	66,086	31,097	77,579	16,132	78,436	22,369
6			74,244	15,721	63,952	9238	82,321		68,961	1951
7				62,993	71,202	20,321	87,289	15,318	83,471	5796
8				62,344			80,773	22,261	80,699	21,478
9				61,640			84,397	20,945		16,245
10				12,681			80,789	12,209		24,937
11							76,135	26,641	74,808	16,594
12									88,721	8588
13									75,941	1990
14										2754
15									80,213	10,950
16									81,595	5404
17									80,424	5584
18									79,487	25,879
19									79,093	25,486
20									77,667	21,966
21									81,417	11,286
22									83,543	25,152
23									82,032	21,587
Mean	78,903		77,561	38,940	70,397	21,811	79,460	19,348	81,480	15,099
± SD	3572		3757	22,443	5038	9751	4631	8502	5602	8595
**Total RE**	**Mean** 78,677 **SD** 6260									

**Table 2 ijms-20-03012-t002:** Total numbers of m^+^RGCs in right (RE) or left (LE) eye retinas.

Retinas	Naive		3 Days		7 Days		14 Days		15 Months	
	RE	LE	RE	LE	RE	LE	RE	LE	RE	LE
1	2434	2201	2135	2062	2163	1678	2034	2409		1994
2	2373	2445	1972	1293	2496	1650	2026	2276		2018
3	2366	2103	2294	1187	1962	1860	2055	2149	2154	1997
4	2362	2249	2547	1682	2262	1971	2242	2425	2297	1987
5	2612	2433	1966	1043	2040	2174	2566	2585	2207	1904
6			2183	1448	2471	2662	2363	2145	2016	1857
7				1719	2612	2746	1950	2661	2267	2019
8				1473			2537	2701	2022	2284
9				1850			2267	1955	2156	1961
10				1411			2559	2660	2196	2004
11							2467	2652	2181	2273
Mean	2358		2183	1453	2287	2106	2279	2420	2166	2027
± SD	144		219	371	247	446	235	257	95	133
**Total RE**	**Mean** 2257 **SD** 229									

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
