# Peer review of "Melanopsin+RGCs Are fully Resistant to NMDA-Induced Excitotoxicity"

_ijms, 2019, doi:10.3390/ijms20123012_

Round 1
Reviewer 1 Report
This paper quantifies the short- and long-term effects of intraocular injection of the excitotoxin NMDA on the melanopsin-containing (m+) and non-melanopsin-containing (Brn3a+) retinal ganglion cells (RGCs) of the rat by means of specific immunolabelling and automated topographic analysis. The authors find that the m+RGCs are unaffected and remain at normal densities, whereas the Brn3a+RGCs suffer a substantial loss of 75% or more.
The study is technically sound, it uses approaches that the group has developed and established over many years. The data are well presented in the text and illustrations, and are adequately discussed in the context of the existing literature. I consider the results reliable. They add to a relatively large body of knowledge about cell type-specific differences in the susceptibility to various kinds of retinal injury and damage, including that to NMDA excitotoxicity. A novel aspect in the present study is the long-term analysis over 15 months post-injection. So the results have some relevance for their research field and may also help as a basis for clinically oriented studies on RGC resilience to injury.
I have no major points of criticism, but a few specific points that should be addressed, plus some typographic and idiomatic corrections.
Specific points
The Abstract states a 75% loss of Brn3a+RGCs at 15 months after NMDA injection (p. 1, l. 30), whereas the text reports a 19% survival (Results Fig. 2) and an 81% loss of Brn3a+RGCs (Discussion p. 12, l. 20). The Abstract needs to be adjusted to match the numbers in the text.
p. 7, Fig. 3 legend: Is the given neighbor radius of 0.0552 mm for the maps of m+RGCs correct? This radius corresponds to a circle area of 0.00957 mm². If that circle contains 16 or more neighbors, as the maps show for dorsal retina, the m+RGC density would be 1671/mm² or more. According to Hannibal et al. 2002, the m+RGC density in dorsal rat retina only is 36-39/mm², so something does not match. Or do I misunderstand the concept of the neighbor maps? Then please clarify in the paper. In previous studies, the group has used a larger neighbor radius of 0.22 mm (ref. 19 = Galindo-Romero et al. 2013; ref. 117 = Vidal-Sanz et al. 2015).
The authors generously cite publications from their own group but give less credit to (earlier) papers from other groups that have dealt with rat m+RGCs / ipRGCs. For example, Hannibal et al. (2002) J Neurosci 22: RC191 (1–7) have reported rat m+RGC densities. The number of rat m+RGCs per retina has been reported by Hattar et al. (2002) Science 295: 1065-1070. The fact that melanopsin antibodies preferentially label rat M1 & M2 ipRGC subtypes (p. 9, l. 17-23) has also been reported by Reifler et al. (2015) Exptl Eye Res 130: 17-28, who used the same melanopsin antiserum as the present study.
Minor points
p. 2, l. 6: Correct to “dendritic”
p. 2, l. 9: Spell out GCL at first appearance. In fact, it appears only once here, so the abbreviation is not needed.
p. 3, l. 10: Change “sensible” to “sensitive”; sensible would mean reasonable.
p. 3, l. 14: Change “important reduction” to “marked reduction” or “strong reduction”. "Important" is a judgment, not a factual description.
p. 5, Fig. 2 legend l. 3: Remove “(d)” and “(m)”, the graph labels don’t have this abbreviation; l. 7: Change to “and experimental retinas”
p. 12, l. 13: Plural “different populations”
p. 12, l. 20: Correct to “… 81% of the Brn3a+RGC …”
p. 12, l. 38: What does “best afford” mean here? Do you mean “best tolerate”?
p. 12, l. 42: Do ipRGCs have axon collaterals in the IPL?
p. 13, l. 26: Correct to “… and were kept …”
p. 13, l. 35: Correct to “… pentobarbital was injected …”
p. 14, l. 9: Delete one “and”
p. 14, l. 11: Put space in “ 3.5-mm posterior”
p. 14, l. 25: The IUPAC nomenclature requires to write NaCl, not ClNa; the cation has to come before the anion.
Ref. 12: Correct to “Dräger” (Umlaut a)
Refs. 15 & 90: Correct to “de Sevilla-Müller” (Umlaut u)
Refs. 24, 25, 26, 111: Correct to “Hallböök” (Umlaut o)
Ref. 50: Correct to “Klöcker” and “Bähr” (Umlaut o & a)
Ref. 74: Correct to “Vidal-Sanz”
Refs. 84 & 86: Correct to “Völgyi” (Umlaut o)
Author Response
Reviewer #1: Specific points
The Abstract states a 75% loss of Brn3a+RGCs at 15 months after NMDA injection (p. 1, l. 30), whereas the text reports a 19% survival (Results Fig. 2) and an 81% loss of Brn3a+RGCs (Discussion p. 12, l. 20). The Abstract needs to be adjusted to match the numbers in the text.
We would like to thank Reviewer #1 for his very constructive criticisms which have helped to improve substantially our manuscript.
We have re-written some sentences in the Abstract (see changes in red) to explain the loss of Brn3a+RGCs, which is progressive up to 7 days but does not progress further. In fact, the statistically significant difference appears between 3 days and 15 months, but not between 7 days and 15 months. (we have also modified Figure 2 accordingly).
p. 7, Fig. 3 legend: Is the given neighbor radius of 0.0552 mm for the maps of m+RGCs correct? This radius corresponds to a circle area of 0.00957 mm². If that circle contains 16 or more neighbors, as the maps show for dorsal retina, the m+RGC density would be 1671/mm² or more. According to Hannibal et al. 2002, the m+RGC density in dorsal rat retina only is 36-39/mm², so something does not match. Or do I misunderstand the concept of the neighbor maps? Then please clarify in the paper. In previous studies, the group has used a larger neighbor radius of 0.22 mm (ref. 19 = Galindo-Romero et al. 2013; ref. 117 = Vidal-Sanz et al. 2015).
The reviewer is right. There was a mistake in the given neighbor radius. We have corrected Fig. 3 legend, indicating that the radius is 0.22 mm.
The authors generously cite publications from their own group but give less credit to (earlier) papers from other groups that have dealt with rat m+RGCs / ipRGCs. For example, Hannibal et al. (2002) J Neurosci 22: RC191 (1–7) have reported rat m+RGC densities. The number of rat m+RGCs per retina has been reported by Hattar et al. (2002) Science 295: 1065-1070. The fact that melanopsin antibodies preferentially label rat M1 & M2 ipRGC subtypes (p. 9, l. 17-23) has also been reported by Reifler et al. (2015) Exptl Eye Res 130: 17-28, who used the same melanopsin antiserum as the present study.
We thank the reviewer for pointing out the previous studies. We have added all these references in the revised version.
In addition, we have noticed that in the previous version of the manuscript the in vivo longitudinal study with OCT included 23 right eyes and 18 left eyes. This was because 5 rats developed a lens cloudiness that hindered proper OCT measurements. For clarity, we have amended the results and included only the 18 rats for which we had data from both eyes. Thus, section 2.3 In vivo SD-OCT mesaurements has been modified (see changes in red), Figure 6 and section 4.3 In vivo measurements of the retinal thickness with SD-OCT, were modified accordingly.
Minor points
p. 2, l. 6: Correct to “dendritic”
Corrected
p. 2, l. 9: Spell out GCL at first appearance. In fact, it appears only once here, so the abbreviation is not needed.
Corrected
p. 3, l. 10: Change “sensible” to “sensitive”; sensible would mean reasonable.
Corrected
p. 3, l. 14: Change “important reduction” to “marked reduction” or “strong reduction”. "Important" is a judgment, not a factual description.
Corrected
p. 5, Fig. 2 legend l. 3: Remove “(d)” and “(m)”, the graph labels don’t have this abbreviation; l. 7: Change to “and experimental retinas”
Corrected. We have modified the Figure and its legend for clarity and to comply with Reviewer #2 request.
p. 12, l. 13: Plural “different populations”
Corrected
p. 12, l. 20: Correct to “… 81% of the Brn3a+RGC …”
Corrected
p. 12, l. 38: What does “best afford” mean here? Do you mean “best tolerate”?
Corrected
p. 12, l. 42: Do ipRGCs have axon collaterals in the IPL?
Corrected. We have eliminated: and axon collaterals
p. 13, l. 26: Correct to “… and were kept …”
Corrected
p. 13, l. 35: Correct to “… pentobarbital was injected …”
Corrected
p. 14, l. 9: Delete one “and”
Corrected
p. 14, l. 11: Put space in “ 3.5-mm posterior”
Corrected
p. 14, l. 25: The IUPAC nomenclature requires to write NaCl, not ClNa; the cation has to come before the anion.
Corrected
Ref. 12: Correct to “Dräger” (Umlaut a)
Corrected
Refs. 15 & 90: Correct to “de Sevilla-Müller” (Umlaut u)
Corrected
Refs. 24, 25, 26, 111: Correct to “Hallböök” (Umlaut o)
Corrected
Ref. 50: Correct to “Klöcker” and “Bähr” (Umlaut o & a)
Corrected
Ref. 74: Correct to “Vidal-Sanz”
Corrected
Refs. 84 & 86: Correct to “Völgyi” (Umlaut o)
Corrected
Reviewer 2 Report
In this manuscript the authors delineate greater susceptibility of Brn3a+ than m+ RGCs to NMDA induced retinal injury. While this has previously been demonstrated, the authors acknowledge and cite the relevant literature and have extended their observations to a longer time period including recovery of m+ RGCs in extended observation. The project is well conceived, the data are solid, the analysis is appropriate, and the discussion is complete. A few specific points follow:
1. Results, p. 5, "There were further reductions at 7 (21,811±9,750 mean±SD, n=6) and 14 days (19,348±8,502 mean±SD, n=10) but these were not statistically significant when compared 1 to 3 days, indicating that in this injury model RGC loss occurs early after NMDA injection but there is no further progression between 3 and 14 days (Figure 2, Table 1). Moreover, at 15 months, the left NMDA4 injected retinas showed significantly lower numbers than their fellow retinas (15,099±8,595 mean±SD, n=23) that corresponded to a survival of approximately 19%, although these values were not different from those obtained at 14 days (Mann Whitney test, p=0,342), indicating that there is no further loss of Brn3a+RGCs between 14 days and 15 months.": While this is correct, ther does appear to be a significant reduction in the values at 3 days and 15 months, suggesting some further effect. The authors might wish to modify this paragraph and the conclusion to reflect this.
2. Table 2.: the LE mean and SD values for 15 months are listed as 'Mean' and '+-SD' instead of the actual values.
Author Response
In this manuscript the authors delineate greater susceptibility of Brn3a+ than m+ RGCs to NMDA induced retinal injury. While this has previously been demonstrated, the authors acknowledge and cite the relevant literature and have extended their observations to a longer time period including recovery of m+ RGCs in extended observation. The project is well conceived, the data are solid, the analysis is appropriate, and the discussion is complete. A few specific points follow:
We would like to thank Reviewer #2 for his very constructive criticisms which have helped to improve substantially our manuscript.
1. Results, p. 5, "There were further reductions at 7 (21,811±9,750 mean±SD, n=6) and 14 days (19,348±8,502 mean±SD, n=10) but these were not statistically significant when compared 1 to 3 days, indicating that in this injury model RGC loss occurs early after NMDA injection but there is no further progression between 3 and 14 days (Figure 2, Table 1). Moreover, at 15 months, the left NMDA4 injected retinas showed significantly lower numbers than their fellow retinas (15,099±8,595 mean±SD, n=23) that corresponded to a survival of approximately 19%, although these values were not different from those obtained at 14 days (Mann Whitney test, p=0,342), indicating that there is no further loss of Brn3a+RGCs between 14 days and 15 months.": While this is correct, ther does appear to be a significant reduction in the values at 3 days and 15 months, suggesting some further effect. The authors might wish to modify this paragraph and the conclusion to reflect this.
We thank the reviewer for pointing out this criticism.
We have re-written some sentences in the Abstract (see changes in red) to explain this fact.
In addition, we have modified the paragraph (see below) and the conclusion (see below) accordingly.
The left NMDA-injected retinas showed significant decreases in the total numbers of Brn3a+RGCs. By 3 days after NMDA injection, the total number of Brn3a+RGCs was 38,940±22,443 (n=9) which is significantly smaller than naïve controls and contralateral retinas (p≤0.001, Mann Whitney test). There were further reductions at 7 (21,811±9,750 mean±SD, n=6) and 14 days (19,348±8,502 mean±SD, n=10) but these were not statistically significant when compared to 3 days, indicating that in this injury model RGC loss occurs early after NMDA injection but there is no further progression between 3 and 14 days (Fig 2, Table 1). Moreover, at 15 months, the left NMDA-injected retinas showed significantly lower numbers than their fellow retinas (15,099±8,595 mean±SD, n=23) that corresponded to a survival of approximately 19%, and these values were significantly smaller than those observed at 3 days (Mann Whitney test, p=0.019), but not from those obtained at 7 days (Mann Whitney test, p=0,187), indicating that there is no further loss of Brn3a+RGCs between 7 days and 15 months. (Figs 1, 2, 3, Table 1).
Conclusions
Intravitreally administered NMDA in adult albino rats: i) induces a massive diffuse loss of Brn3+RGCs which is evident within the first week and does not progress further; ii) Causes a thinning of the inner retina by 3 months that further progresses up to 15 months; iii) Triggers a transient downregulation of melanopsin expression, that is evident at 3 days and recovers fully by 14 days, and; iv) Does not induce m+RGCs loss.
2. Table 2.: the LE mean and SD values for 15 months are listed as 'Mean' and '+-SD' instead of the actual values.
There was an error in the editing of this Table which we have corrected.
In addition, we have noticed that in the previous version of the manuscript the in vivo longitudinal study with OCT included 23 right eyes and 18 left eyes. This was because 5 rats developed a lens cloudiness that hindered proper OCT measurements. For clarity, we have amended the results and included only the 18 rats for which we had data from both eyes. Thus, section 2.3 In vivo SD-OCT mesaurements has been modified (see changes in red), Figure 6 and section 4.3 In vivo measurements of the retinal thickness with SD-OCT, were modified accordingly.